# The Effect of a 2-Week Red Ginseng Supplementation on Food Efficiency and Energy Metabolism in Mice

**DOI:** 10.3390/nu12061726

**Published:** 2020-06-09

**Authors:** Hyejung Hwang, Jisu Kim, Kiwon Lim

**Affiliations:** 1Physical Activity and Performance Institute (PAPI), Konkuk University, 120 Neungdong-ro, Gwangjin-gu, Seoul 05029, Korea; hfilm@konkuk.ac.kr (H.H.); kimpro@konkuk.ac.kr (J.K.); 2Department of Physical Education, Konkuk University, 120 Neungdong-ro, Gwangjin-gu, Seoul 05029, Korea

**Keywords:** red ginseng, body weight, energy metabolism, FAT/CD36, GLUT4

## Abstract

Red ginseng (RG) ingestion reportedly affects body weight, food intake, and fat accumulation reduction. It also induces changes in energy metabolism regulation and glycemic control. Previously, 2-week RG ingestion with endurance training was found to enhance fat oxidation during exercise. However, such effects on energy metabolism and the expression of mRNAs related to energy substrate utilization in resting mice (untrained mice) are still unclear. Here, we determined the effect of RG on energy metabolism and substrate utilization in untrained male mice. Twenty-four mice were separated into an RG group that received a daily dosage of 1 g/kg RG for 2 weeks, and a control (CON). Energy expenditure, blood and tissue glycogen levels, and expression of mRNAs related to energy substrate utilization in muscles were measured before and 2 weeks after treatment. Total food intake was significantly lower in the RG than in the CON group (*p* < 0.05), but final body weights did not differ. Carbohydrate and fat oxidation over 24 h did not change in either group. There were no significant differences in gastrocnemius *GLUT4*, *MCT1*, *MCT4*, *FAT/CD36*, and *CPT1b* mRNA levels between groups. Thus, the effects of RG ingested during rest differ from the effects of RG ingestion in combination with endurance exercise; administering RG to untrained mice for 2 weeks did not change body weight and energy metabolism. Therefore, future studies should consider examining the RG ingestion period and dosage for body weight control and improving energy metabolism.

## 1. Introduction

Panax ginseng possesses 10 biologically active steroidal glycosides, including ginsenosides, which have been generally recognized as a major active component of Korean red ginseng (RG) [1]. Depending on the composition ratio of ginsenosides and structural and physiological differences, the ginsenosides in RG can affect the human body in different ways. RG has been shown to have anti-stress, anti-fatigue, anti-cancer, hypotensive, and antioxidant properties [2,3,4,5]

Recently, some researchers have shown, using an animal model, that RG ingestion might reduce body weight and fat accumulation under a high-fat diet [6,7]. Kim et al. [6] reported that the ingestion of RG saponin solution (200 mg/kg) for 3 weeks leads to reductions in body weight, food intake, and adipose tissues in obese mice induced by high-fat diet feeding for 5 weeks. In addition, daily RG intake for 12 weeks improves insulin signaling by enhancing the phosphorylation of IR-b, IRS-1, Akt, and GSK3a and increases GLUT4 translocation in skeletal muscle tissue [8]. Moreover, we previously reported that 2 weeks of RG ingestion combined with endurance training increases fat oxidation and glycogen-sparing effects during exercise [9]. Our previous study examined the effect of 2 weeks of combined RG ingestion and endurance exercise; however, the effect of the same RG dose and ingestion periods without endurance exercise is still unknown. Furthermore, most previous studies have investigated the effects of RG on blood profiles, hormones, and insulin sensitivity, and the effects of RG related to energy metabolism are still unclear. Hence, it is considered that the effect of RG dose on fat oxidation and the expression of mRNAs related to energy metabolism should be evaluated. Therefore, the purpose of the present study was to determine the effect of oral RG ingestion for 2 weeks on untrained male mice through a comprehensive analysis of resting energy metabolism, performed by measuring blood and tissue glycogen and assessing the expression of relevant mRNAs in muscle tissue. Our study showed that the effects of RG ingested during resting conditions differ from those when RG ingestion is combined with exercise and that the administration of RG for 2 weeks did not effectively control body weight and energy metabolism.

## 2. Materials and Methods

### 2.1. Animals and Treatment

Male ICR mice (*n* = 24) of 7 weeks of age were used from Orient Bio Inc. (Seongnam, Korea). All mice were individually housed in standard plastic cages under controlled conditions of 50% humidity and 23 ± 1 °C, with 12-h light (from 08:00 to 20:00) and darkness (from 20:00 to 08:00) cycles. Mice were allowed to adapt to laboratory housing conditions for 1 week. Twenty-four male mice were separated into two groups, namely CON (no RG administered, *n* = 12) and RG (RG administered, *n* = 12). The mice were given water and a standard chow diet (5L79, Orient Bio Inc., Seongnam, Korea) ad libitum. Every day at 9:00, the body weight and food intake of each mouse was measured using a balance (Innotem IB-6100, Korea). This study was conducted in accordance with the ethical guidelines of the Konkuk University Institutional Animal Care and Use Committee. The approval number is “KU10040”.

### 2.2. RG Extract

RG extract (Korea Ginseng Corp., Seoul, Korea) dissolved in distilled water was administered via oral gavage daily to the RG group at a dose of 1 g/kg (19.64 mg ginsenosides/g RG extract) for 2 weeks. The CON group was orally gavaged with 5 mL/kg distilled water daily for 2 weeks. The mice in the CON group were given the same amount of distilled water.

### 2.3. Energy Expenditure

The energy expenditure (including resting metabolic rate (RMR), CHO, and fat oxidation) was measured pre and 2 weeks post RG administration. The measure for respiratory gas was assessed using a specially manufactured mouse metabolic chamber, and O_2_ uptake and CO_2_ production were measured using a mass spectrometer (RL-600, Arco system, Chiba, Japan). The gas sampling rate was set to 1 min in each chamber. Respiratory gas was measured using open-circuit methods. Flow velocity within the chamber was set to 1.2 L/min for 24 h (H.J. Hwang et al., 2013, H. Hwang et al., 2014). All mice were given free access to water and food during the 24-h measurement period. RMR was calculated by extrapolating the mean of 12-point data after 1 h in light within 24 h. In addition, the metabolism of the active period was estimated based on the 12-h dark phase and the inactive period was estimated based on the 12-h light phase. The oxidation of carbohydrates and fats was calculated using VO_2_ and VCO_2_ measurements based on the following equations:CHO oxidation (g/min) = 4.210 VCO_2_ − 2.962 VO_2_(1)
FAT oxidation (g/min) = 1.695 VO_2_ − 1.701 VCO_2_(2)

### 2.4. Blood Analysis

Blood samples were collected using EDTA tubes (BD Vacutainer K2 EDTA 10.8 mg, Plus Blood Collection Tubes, USA) and centrifuged at 3000 rpm at 4 °C for 15 min. Plasma glucose was analyzed using commercial kits (Asan Pharmaceutical Co., Seoul, Korea). Plasma insulin levels were measured using an enzyme-linked immunosorbent assay kit (Morinaga Bioscience Laboratory, Yokohama, Japan), and the levels of plasma-free fatty acids (FFAs) were analyzed using a non-esterified fatty acid kit (LabAssay™ NEFA kits, Wako Pure Chemical Industries, Osaka, Japan).

### 2.5. Glycogen Analysis

Glycogen was measured using the amyloglucosidase method established in a previous study (H. Hwang et al., 2014). Approximately 30 mg of liver and muscle tissue was placed on ice. We added 0.5 mL of 2 N HCl and incubated the samples for 2 h at 96 °C after adding 1.5 mL of 0.67 M NaOH. The supernatant, a 100-µL sample of glucose, was taken, and 1 mL of reaction buffer was added to it; the solution was incubated for 30 min at 37 °C. Next, the sample reactions were stopped by putting them on ice. The glucose content level was analyzed in a 96-well plate using a spectrophotometer (Multiskan Go microplate reader, ThermoFisher, Waltham, MA, USA) at an absorbance of 340 nm.

### 2.6. RT-PCR (Reverse Transcriptase PCR) Analysis

Skeletal muscle gene expression was analyzed by semi-quantitative analysis using RT-PCR. Total RNA was extracted using Trizol (Ambion Life technologies, Carlsbad, CA, USA) according to the manufacturer’s instructions. Briefly, gastrocnemius muscle tissue was homogenized (Qiagen, TissueRuptor, Hilden, Germany) with 1 mL of Trizol reagent. After 5 min of incubation at room temperature, 200 µL of chloroform was added to the tubes, which were then centrifuged for 15 min at 12,000× *g*. The supernatant was transferred to another tube and isopropanol was added (Iso-Propyl Alcohol, Duksan Pure Chemicals, Asan, Korea). RNA pellets were diluted in 30 µL of DEPC water and heated at 55 °C for 10 min. The RNA was quantified by measuring the absorbance at 260 nm using a Nano Block (Multiskan Go microplate reader, ThermoFisher, Waltham, MA, USA), and the purity of the RNA was assessed based on the 260/280-nm ratio. RNA was extracted, and cDNA synthesis was performed using the cDNA Synthesis Master Mix, according to the manufacturer’s instructions (GenDEPOT, Barker, TX, USA). The synthesized cDNA was stored at −20 °C. 

Gene expression was analyzed using amfiEco *Taq* DNA Polymerase, according to the manufacturer’s instructions (GenDEPOT, Barker, TX, USA). A 1-µL aliquot of cDNA was mixed with 24 µL of 2× reaction buffer, PCR water, *Taq* DNA polymerase, and forward and reverse primers to obtain a final volume of 25 µL. PCR was carried out using the Thermal Cycler Dice Touch (TAKARA BIO Inc., SHIGA, Japan). The primer sequences were designed using information from GenBank, a public database of the National Center for Biotechnology Information. The primer sequences, annealing temperatures, and product lengths are presented in Table 1. The conditions of qRT-PCR cycling for each gene are as follows: *GAPDH*: 52 °C, 18 cycles; *GLUT4*: 52 °C, 23 cycles; *MCT1*: 52 °C, 25 cycles; *MCT4*: 52 °C, 22 cycles; *FAT/CD36*: 53 °C, 24 cycles; *CPT1b*: 53 °C, 24 cycles. The PCR products were run on a 10% agarose (Sigma Aldrich, Steinheim, Germany) gel with 1× TAE (tris-acetate-EDTA) buffer and Safe-Pinky DNA gel staining solution (10,000×; GenDEPOT, Barker, TX, USA). PCR product bands were measured using Print Graph 2M (ATTO, Sungnam, Korea). We normalized expression using *GAPDH* as a house-keeping gene, and results from the RG groups were expressed as fold change based on levels in the CON group in percent (%). The PCR products were run on a 10% agarose (Sigma Aldrich, Steinheim, Germany) gel with 1× TAE buffer and Safe-Pinky DNA gel staining solution (10,000×; GenDEPOT, Barker, TX, USA). PCR product bands were measured using Print Graph 2M (ATTO, Sungnam, Korea).

### 2.7. Statistical Analysis

Results are shown as the mean ± standard error (SE). The statistical analysis was performed using the SPSS 19.0 program (SPSS Inc., Chicago, IL, USA). The differences between groups were analyzed using an unpaired *t*-test. The effects of time and group allocation were analyzed using a two-way repeated-measures analysis of variance. Differences were considered significant at *p* < 0.05.

## 3. Results

### 3.1. Body Weight, Food Intake, and Abdominal Tissue Weight

Changes in body weight, food intake, and abdominal adipose tissue weights before and after the 2-week experimental period are shown in Table 2. Initial body weight, final body weight, and body weight gain did not differ between the two groups. However, the total food intake (RG: 85.95 ± 0.96, CON: 96.23 ± 2.67 (g/2 weeks), RG: 275.91 ± 3.23 CON: 308.92 ± 8.96 (kcal/2 weeks)) and daily food intake (RG: 5.37 ± 0.06, CON: 6.01 ± 0.16 (g/day), RG: 17.24 ± 0.20 CON: 19.31 ± 0.56 (kcal/day)) were significantly lower in the RG group than in the CON group; adipose tissue and total tissue measurements did not differ between the CON and RG groups.

### 3.2. Oxygen Uptake, CO_2_ Production, and RMR

As shown in Figure 1A,B, the sum of 24-h oxygen uptake and CO_2_ production before and after the 2-week experiments was not significantly different between the CON and RG groups. After 2 weeks of RG intake, there was no significant difference between the CON and RG groups in the comparison of RMR. Moreover, energy expenditure in the active period over 12 h was not significantly different between the two groups (Table 3).

### 3.3. Carbohydrate and Fat Oxidation over 24 h

The changes in carbohydrate and fat oxidation occurring 24 h after 2 weeks of RG intake are presented in Figure 2A,B. There was no difference found in the 24-h sum of carbohydrate and fat oxidation between the two groups.

### 3.4. Plasma Parameters

The plasma concentrations of glucose, insulin, and FFAs are shown in Table 4. The plasma parameters did not differ significantly between the CON and RG groups.

### 3.5. Glycogen Concentrations

The glycogen concentrations in the CON and RG groups are presented in Table 5. There were no differences in tissue (liver and muscle) glycogen concentrations between the two groups.

### 3.6. Expression of Skeletal Muscle mRNAs Related to Energy Metabolism

Changes in *GLUT4*, *MCT1*, *MCT4*, *FAT/CD36*, and *CPT1b* mRNA expression in skeletal muscle (gastrocnemius) after the 2-week experimental period are shown in Figure 3. There was no significant difference in mRNA expression between the two groups.

## 4. Discussion

This study involved a comprehensive analysis of the effect of RG ingestion for 2 weeks on male mice. The main results of this study are as follows. First, we found that the total amount of food intake was significantly lower in the RG group than in the CON group, whereas mean body weight did not differ between groups. Second, neither oxygen uptake nor oxidation of carbohydrates and fat differed between the groups. Third, there was no difference in the expression of mRNAs related to gastrocnemius energy substrate utilization between the two groups.

### 4.1. Effect of RG on Total Food Intake and Mean Body Weight

A number of researchers have published studies on RG administration in relation to anti-obesity effects, indicating that RG treatments reduce both food consumption and body weight [10,11,12]. The oral ingestion of RG (200 mg/kg) for 18 weeks in combination with a high-fat diet significantly decreases food intake and increases serum blood leptin levels [13]. Moreover, the ingestion of fermented RG, levan, and their combination for 11 weeks resulted in significant effects, reducing total food intake and total body weight [14]. Additionally, RG treatment in rats increases insulin sensitivity and inhibits hypothalamic neuropeptide Y [15], which mediates many leptin-induced effects of food intake [13]. In our previous study [9], treatment with 1 g/kg RG for 2 weeks (the same dosage used in the present study) combined with endurance training increased food intake, whereas this increase was not observed in the CON group.

The present results support these prior results only in terms of food consumption reduction. Here, the total amount of food intake was significantly lower in the RG group than in the CON group (*p* < 0.05), but the mean body weight did not differ between the groups. This partially aligned with the results of the study conducted by Park et al. [16], which suggested that RG administered at doses of 500, 1000, and 2000 mg/kg per day for 4 weeks causes no significant change in body weight and dietary food intake. Unfortunately, our study did not investigate leptin levels. Still, based on previous studies and the findings of this study, we assumed that RG intake over 2 weeks might affect dietary food intake by regulating leptin levels. However, the extent of the reduction in body weight could vary depending on the intake period and dose.

### 4.2. Effect on Oxygen Uptake and Oxidation of Carbohydrates and Fats

Xiao et al. [17] measured energy metabolism over a 24-h period in rats that were administered RG (1.66 g/kg) for 20 days and proposed that, although oxygen consumption, heat production, and energy expenditure increase upon RG administration during the daytime, there are no differences in energy metabolism in nocturnal rats administered RG at night. Furthermore, it has been reported that the expression level of hepatic AMPK, which is involved in regulating fat metabolism, also increases upon RG intake. Similarly, the administration of 2 mg of Panax red ginseng (PRG) extract (which includes 34.41 mg/g of ginsenosides) resulted in higher O_2_ consumption, and it was suggested that ginseng treatment increases the energy expenditure of mice with diet-induced obesity. Moreover, PRG ingestion activated lipolysis in white adipose tissues and promoted energy expenditure in brown adipose tissues [18].

In our experiment, oxygen uptake and carbohydrate and fat oxidation over 24 h, both before and 2 weeks after treatment, did not differ between the two groups. Likewise, there was no significant difference in RMR or carbohydrate and fat oxidation after 24 h. According to the results of our prior experiment, the ingestion of RG (1 g/kg) with endurance training promotes fat oxidation during exercise. However, the ingestion of RG without training did not induce changes related to resting energy metabolism. We had inferred based on our previous study that endurance training attenuated the effect of RG ingestion on resting energy metabolism, and the results of this experiment confirmed that RG ingestion for 2 weeks did not affect resting energy metabolism.

Several studies have reported an association not only between the ingestion of Panax ginseng or RG and changes in energy metabolism regulation but also between Panax ginseng or RG ingestion and glycemic control [12,19,20,21,22]. Karunasagara et al. [23] reported that the administration of RG (60 mg/kg) for 4 weeks can prevent diabetic kidney disease by decreasing the levels of metabolic and blood parameters. Additionally, 8-week administration of the non-saponin fraction of KRG (KGC05P0) was found to regulate the levels of fasting glucose, glucose tolerance, insulin, and other cytokines in the blood of diabetic mice. Furthermore, GLUT2 expression was significantly increased in the liver upon administration of both medium (200 mg/kg) and high (400 mg/kg) doses of RG [16]. Li et al. [22] investigated the glucose-control effects of RG ingestion using a high-fat-diet-fed animal model and reported that the ginsenoside Rg1, a component of RG, promotes the plasma membrane translocation of GLUT4 in C2C12 skeletal muscle cells, which contributes to a reduction in blood glucose levels through the transportation of circulating glucose.

In contrast to these prior findings, the results of our study did not provide evidence supporting the anti-diabetic effect of RG or ginsenoside administration on blood glucose regulation in mice. We found no difference in the amount of glycogen stored in the liver and muscle after 2 weeks of RG administration and no changes in any blood parameter. However, since we did not conduct an oral glucose tolerance test, we could not examine the insulin response. Furthermore, the previously reported findings discussed here are from studies that investigated the long-term ingestion of RG using a high-fat-diet-fed animal model, and these studies also differ from the present study in terms of the ingestion period and dosage. The content of ginsenosides and other components of RG also differed between the present study and previous ones. In Korea, the permissible daily dose of ginsenosides, such as Rg1 and Rb1, ranges from 3 to 80 mg [24]. The dose used in this study is the normal dose (19.64 mg of ginsenosides) within this range, and, when assessing the previous study, a high dose was needed to induce a change in energy metabolism at rest.

One limitation of this study is that, for animal studies, there is no well-established protocol to measure whole-body energy metabolism for 24 h. However, measuring changes in energy metabolism is an essential part of obesity studies that examine energy substrate utilization, and these studies are necessary to prevent obesity and promote health. Therefore, it is important to develop reliable methods of energy metabolism measurement for animal studies.

### 4.3. Effect on Expression of mRNAs Related to Energy Metabolism

Monocarboxylate transporters (*MCTs*) are expressed inside and outside working muscle cells and enable lactate translocation during exercise [25]. Understanding how lactate substrates function can improve our knowledge of carbohydrate metabolism. Therefore, we measured the expression of *MCT1* and *MCT4* mRNA in the gastrocnemius. Our results showed no difference in the expression of these mRNAs between the CON and RG groups. During exercise, long-chain fatty acids are the primary energy source and are imported into skeletal muscle mitochondria with the help of fatty acid translocase/CD36 *(FAT/CD36*), which might interact with carnitine palmitoyltransferase 1 (*CPT1*) [26]. In our previous study, the expression of CPT1b in skeletal muscle increased after 2 weeks of RG ingestion combined with endurance training. However, in the present study, the ingestion of the same amount of RG without exercise did not affect the expression of mRNAs related to fatty acid translocation in skeletal muscle. Hence, our findings indicate that, although the ingestion of RG might effectively regulate factors related to fat oxidation when combined with exercise and a high-fat diet, RG administered to resting mice does not affect the control of energy metabolism.

## 5. Conclusions

In contrast to previous studies, there was no change in body weight, plasma insulin, or expression of mRNAs related to carbohydrate and fat metabolism after RG intake in this study. Differences in the duration of RG intake and dosage might serve as possible reasons for these discrepancies. In conclusion, the effects of RG intake vary depending on the dosage, duration of intake, and whether intake is combined with exercise. Future studies should thus examine the long-term effects of RG ingestion and dosage in terms of body weight control and energy metabolism.

## Figures and Tables

**Figure 1 nutrients-12-01726-f001:**
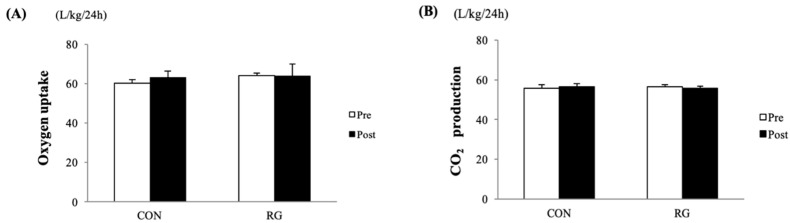
Comparison of the sum of oxygen uptake (**A**) and CO_2_ production (**B**) for 24 h between the two groups before and after 2 weeks of red ginseng (RG) intake. Values are presented as the mean ± SE.

**Figure 2 nutrients-12-01726-f002:**
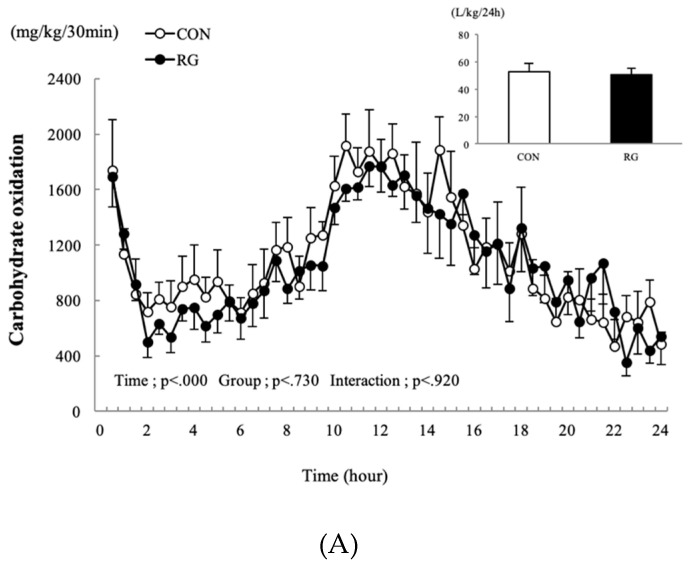
Changes in carbohydrate oxidation (**A**) and fat oxidation (**B**) after 2 weeks of red ginseng (RG) intake, and the sum for the control (CON) and RG groups 24 h after 2 weeks of RG intake. Values are presented as the mean ± SE.

**Figure 3 nutrients-12-01726-f003:**
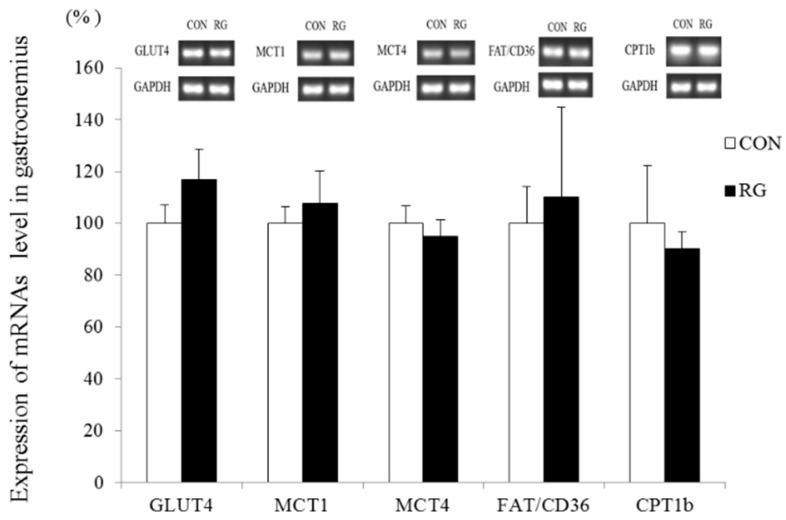
Reverse transcription PCR analysis of *GLUT4*, *MCT1*, *MCT4*, *FAT/CD36*, and *CPT1b* mRNA levels in the gastrocnemius after 2 weeks of red ginseng (RG) intake. Values are presented as the mean ± SE.

**Table 1 nutrients-12-01726-t001:** Primer sequences, annealing temperatures, and product lengths.

Gene	Accession Number	Sequences	Annealing Temperature (°C)	Product Length (bp)
*GAPDH*	BC145810.1	F-5′-aactttggcattgtg gaagg–3′ R-5′-acacattgggggtaggaaca–3′	52	215
*GLUT4*	AB008453.1	F-5′-ttctggctctcacagtactc–3′ R–5′-cattgatgcctgagagctgt–3′	52	300
*MCT1*	NM_009196.4	F-5′-gctggaggtcctatcagcag–3′ R-5′-agttgaaagcaagcccaaga–3′	52	172
*MCT4*	NM_001038654	F-5′-acggctggtttcataacagg-3′ R-5′-ccaatggcactggagaactt-3′	52	233
*FAT/CD 36*	NM_007643.4	F-5′-ggccaagctattgcgacat-3′ R-5′-cagatccgaacacagcgtaga-3′	53	129
*CPT1b*	NM_009948.2	F-5′-atcatgtatcgccgcaaact-3′ R-5′-ccatctggtaggagcacatgg-3′	53	85

**Table 2 nutrients-12-01726-t002:** Changes in body weight, food intake, and abdominal adipose tissue.

	CON	RG
Initial body weight (g)	35.79 ± 0.31	35.87 ± 0.56
Final body weight (g)	36.10 ± 0.38	36.80 ± 0.58
Body weight gain (g)	0.31 ± 0.29	0.93 ± 0.30
Total amount of food intake (g/2 weeks)	96.23 ± 2.67	85.95 ± 0.96 **
Total amount of food intake (kcal/2 weeks)	308.92 ± 8.96	275.91 ± 3.23 **
Daily food intake (g)	6.01 ± 0.16	5.37 ± 0.06 **
Daily food intake (kcal)	19.31 ± 0.56	17.24 ± 0.20 **
Epididymal (g)	0.71 ± 0.05	0.86 ± 0.09
Perirenal (g)	0.32 ± 0.03	0.38 ± 0.05
Mesenteric (g)	0.68 ± 0.02	0.66 ± 0.03
Total adipose tissue (g)	1.70 ± 0.01	1.90 ± 0.03

Changes in the body weight, food intake, and abdominal adipose tissue weight in the CON and RG groups. Values are given as the mean ± SE. ** vs. CON, *p* < 0.01.

**Table 3 nutrients-12-01726-t003:** Resting metabolic rate and active period (dark phase) after 2 weeks of red ginseng (RG) intake.

	CON	RG
RMR (kcal/day)	8.42 ± 0.57	7.77 ± 0.55
Active period (kcal/12 h)	4.26 ± 0.24	4.00 ± 0.36
Inactive period (kcal/12 h)	4.52 ± 0.25	4.44 ± 0.37

Values are presented as the mean ± SE.

**Table 4 nutrients-12-01726-t004:** Levels of glucose, insulin, and free fatty acids (FFAs) in plasma after 2 weeks of red ginseng (RG) intake.

	CON	RG
Glucose (mg/dL)	159.61 ± 11.96	164.10 ± 10.39
Insulin (ng/mg)	0.91 ± 0.09	1.03 ± 0.16
FFA (mEq/L)	0.46 ± 0.06	0.38 ± 0.03

Values are presented as the mean ± SE.

**Table 5 nutrients-12-01726-t005:** Changes in the glycogen concentration (µmol/g) in the liver, gastrocnemius white muscle, and gastrocnemius red muscle after 2 weeks of red ginseng (RG) intake.

	CON	RG
Liver	343.1 ± 12.2	330.2 ± 20.1
Gastrocnemius white muscle	29.7 ± 2.3	31.4 ± 2.2
Gastrocnemius red muscle	17.8 ± 2.3	16.7 ± 0.5

Values are presented as the mean ± SE.

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
