# Peer review of "The Effect of a 2-Week Red Ginseng Supplementation on Food Efficiency and Energy Metabolism in Mice"

_nutrients, 2020, doi:10.3390/nu12061726_

Round 1
Reviewer 1 Report
This study examined the effects of red ginseng supplementation on body weight, food intake, and energy metabolism. The authors found that red ginseng supplementation in the drinking water reduced food intake but with no effects on body weight. Interestingly, there is also no notable differences in metabolic rate, thus the acute anorectic effects of red ginseng is not fully known. The fact that there are no differences in body weight is not surprising since supplementation was delivered only for 2 weeks; it would be of interest to determine in future studies whether long-term supplementation with red ginseng would lead to a gradual reduction in body weight. The paper is well-constructed and the authors have provided a thoughtful discussion.
I have a few points to clarify below:
- Mice supplemented with red ginseng have slightly elevated blood glucose levels than control mice. This might be related to enhanced glycogen mobilization from the liver, as the authors show lower liver glycogen levels. In addition to skeletal muscle data in Figure 2, I wonder if gene expression studies using liver tissue would reveal a potential explanation for the effects of red ginseng.
- Please clarify whether red ginseng was administered to the animals via drinking water (Ad lib drinking) or oral gavage. If ad lib drinking, how do you ensure that the animals consumed the correct dosage? If oral gavage, please indicate this in the methdos.
- Were animals single-housed during the 2-week supplementation period?
- Please clarify the unit for food intake. It should be in kcal
Author Response
Response to Reviewer 1 Comments
Thank you very much for your constructive comments and suggestions. We have considered all the comments carefully and revised our manuscript accordingly. Our responses to each comment (Shown in red) are as follows.
Point 1: Mice supplemented with red ginseng have slightly elevated blood glucose levels than control mice. This might be related to enhanced glycogen mobilization from the liver, as the authors show lower liver glycogen levels. In addition to skeletal muscle data in Figure 2, I wonder if gene expression studies using liver tissue would reveal a potential explanation for the effects of red ginseng.
Response 1: Thank you for your constructive feedback
1) At first, there was no significant difference between the two groups in blood and liver glycogen concentrations. We revised the significant difference-maker (*) in Table 4 and 5.
The blood glucose levels tended to be higher in the RG group, and liver glycogens tended to be lower. It also demonstrated the potential to affect the mobilization of liver glycogen. In our previous study, RG ingestion with endurance training for two weeks, we found the saving effect in the liver and white muscle glycogen use after 1 h exercise. Therefore, we agree with your opinion that the ingestion of RG may have an effect on CHO metabolism. However, there was no significant difference in the liver and muscle glycogen concentrations and blood parameters, therefore we did not mention it in the discussion part section.
2) This study did not examine the gene expression in the liver, but there are many studies showing that RG ingestion is effective in liver disease (especially “fatty liver”).
In this study, if we find that significant changes of the ingestion of RG in energy metabolism and blood variables, we believe that gene expression tests related to the use of energy metabolism substrates in the liver will be meaningful data.
Point 2: Please clarify whether red ginseng was administered to the animals via drinking water (Ad lib drinking) or oral gavage. If ad lib drinking, how do you ensure that the animals consumed the correct dosage? If oral gavage, please indicate this in the methdos.
Response 2: We have modified the animal treatment methods in the article with more details provided (Lines 92-95).
Point 3: Were animals single-housed during the 2-week supplementation period?
Please clarify the unit for food intake. It should be in kcal
Response 3: During the 2 wks experiment period, animals were single-housed and measure to food intake and weight were measured at 09:00 am daily. We have added the units of food intake in Table 2 (Lines 81-88, 188-189).

Reviewer 2 Report
Nutrients-803405: Hwang HJ et al. Effect of a two-week red ginseng supplementation on food efficiency and resting energy metabolism in mice.
The authors wanted to determine the effect of red ginseng (RG) on energy metabolism and substrate utilization of untrained male mice. Twenty-four mice were separated into two groups, RG and CON (control), with n=12 each. The RG group was dosed daily for 2 wk with 1 g/kg RG. Resting metabolic rate, blood and tissue glycogen level, and the expression of mRNAs related to energy substrate utilization in muscles were measured before and 2 wk after treatment. Total amount of food intake was significantly lower in RG vs. CON whereas all the other parameters did not differ significantly after two weeks. The authors concluded that the effects of RG ingested during resting conditions differed from those of RG ingested in combination with exercise, and that administering RG to untrained mice for 2 wk did not effectively control body weight and energy metabolism.
Overall, this is a well designed study and a well-written manuscript. However, I have some major concerns regarding calculating and interpreting the calorimetric data. The authors report throughout the manuscript about resting energy metabolism. This term is misleading. For my understanding they report about 24h total energy expenditure (TEE) without an additional training program in contrast to an earlier study with an additional training program published also in Nutrients in 2014. TEE is the sum of 1) resting energy ependiture (REE) or resting metabolic rate (RMR), 2) diet-induced thermogenesis (DIT), and 3) exercise activity thermogenesis (EAT). Mice and rats never rest over 24h. Instead there is a highly increased EAT during night time. The authors did not report about the sampling rate of CO2 production and O2 consumption. Instead, they report just about integrated 30 min intervals. If there is a sampling rate of, for example, one data point per minute, one could easily calculatate RMR by just taking the 10 or 15 minimum values of energy expenditure within 24h and extrapolating the mean of these data to 24h. By using this approach, one can really differentiate between changes in RMR and DIT/EAT and also the corresponding substrate oxidation rates. The authors should then show not only the time courses for carbohydrate / fat oxidation but also energy expenditure. The same could be done retrospectively with the data published in Nutrients in 2014. Possibly, there are no changes in TEE but between the ratio of RMR and DIT / EAT.
Furthermore, how exactly did the authors assess food intake? Did the authors also differentiate between energy intake by food and energy excretion by Faeces?
Minor: In the figures, carbohydrate and fat oxidation rates should be given in mg/kg/30 min, not as ml/kg/30 min
Author Response
Response to Reviewer 2 Comments
Thank you very much for your constructive comments and suggestions. We have considered all the comments carefully and revised our manuscript accordingly. Our responses to each comment (shown in red) are as follows.
Point 1: This term is misleading. For my understanding they report about 24h total energy expenditure (TEE) without an additional training program in contrast to an earlier study with an additional training program published also in Nutrients in 2014. TEE is the sum of 1) resting energy ependiture (REE) or resting metabolic rate (RMR), 2) diet-induced thermogenesis (DIT), and 3) exercise activity thermogenesis (EAT). Mice and rats never rest over 24h. Instead there is a highly increased EAT during night time. The authors did not report about the sampling rate of CO2 production and O2 consumption. Instead, they report just about integrated 30 min intervals. If there is a sampling rate of, for example, one data point per minute, one could easily calculatate RMR by just taking the 10 or 15 minimum values of energy expenditure within 24h and extrapolating the mean of these data to 24h. By using this approach, one can really differentiate between changes in RMR and DIT/EAT and also the corresponding substrate oxidation rates. The authors should then show not only the time courses for carbohydrate / fat oxidation but also energy expenditure. The same could be done retrospectively with the data published in Nutrients in 2014. Possibly, there are no changes in TEE but between the ratio of RMR and DIT / EAT.
Response 1:
1) The O2 consumption and CO2 production data figures have been added following to your comments. (please see Figure 1)
2) The energy expenditure measurements (included RMR) have been added to the Materials and Methods part of the paper which provides more detailed explanation and correspondingly, has been revised in the article. (Line 96-106, Section 2.3.)
3) Based on your suggestion, we calculated the RMR from the 24-hour respiratory gas data. Also, we calculated energy expenditure (kcal) of mice active period using the dark phase of 12h. However, there was no significant difference between the two groups in RMR and the active period after RG intake. (please see Table 3, Line 196-200)
4) We cautiously assume the following: in this study, all animals were provided with an ad libitum that they are free access the feed at any time during measurement. Therefore, it is difficult to clearly distinguish the DIT response.
5) The energy expenditure of each carbohydrate and fat oxidation amount is expressed in the unit of kcal. We have added this issue to the results section.
Point 2: Furthermore, how exactly did the authors assess food intake? Did the authors also differentiate between energy intake by food and energy excretion by Faeces?
Response 2:
1) Every day, at 09:00 at body weight and food intake of each mouse was measured by balance (Innotem IB-6100, Korea) (Line 86-88).
2) The food intake assessed was based on the amount of feed on the day before and the remaining amount of feed on the day after each mouse. However, in our experiments, energy excretion through the faeces of the mouse was not evaluated separately.
Point 3: Minor: In the figures, carbohydrate and fat oxidation rates should be given in mg/kg/30 min, not as ml/kg/30 min
Response 3: We have modified Figure 2 (A) and (B).

Reviewer 3 Report
The paper descirbes some metabolic effects of RG admisinitration in mice. The introduction provides a good background but the fact that this study is not looking at effects in sedentary vs excercied mice, but rather loosely compared to a previous study that used excericed mice is not clear. The methods lack some important information. The results are presented well, but why is RMR discussed in the methods but not presented as a result? Additionally, there are several mentions in the discussion of results that were never shown. The conclusion focuses much more heavily on other studies than the one in the paper. Referencing style needs to be fixed for consistency.
[Abstract] Should be some statement as to signficiance. What is red ginseng and why is it worth looking at?
[Line 20] Should say earlier what you are comparing to in terms of excercie - mention previous study.
[Lines 20-21] Differed in what way?
[Lines 22-23] Needs supportive arguments. If short-term administration had no effects, why bother looking at long term?
[Lines 34-35] In what organism? Human? Mouse? Where are the references?
[Lines 37-38] Need to check causality of the statement. Are you saying that the RG or the high-fat diet improved insulin sensitivity?
[Section 2.1] What kind of diet? Standard chow? High-fat?
[Section 2.2] How were they administered? By gavage? Where the volumes of RG and control similar? (what was the concentration of RG dissolved in water)
[Section 2.3] How was RMR measured/calculated? Was the pre-2wk RMR experimental setup the same? Are there references for the fat and carbohydrate oxidation calculations?
[Section 2.6] What tissue was used? What were the concentrations of Tag and primers? What were the PCR cycling conditions? What calculation was used - was the RG expressed as fold-change compared to control? Was a house-keeping gene used to normalise expression?
[Table 2] What does the single asterix footnote refer to?
[Tables 3 and 4] Is only the one value shown as mean ± SE?
[Line 163] Oxygen uptake is not reported in the results
[Line 185] Did you measure leptin?
[Line 190-191] what time of day/night was the RG administered in this study?
[Line 198-199] Oxidation was not reported before the 2 weeks.
Author Response
Response to Reviewer 3 Comments
Thank you very much for your constructive comments and suggestions. We have considered all the comments carefully and revised our manuscript accordingly. Our responses to each comment (shown in red) are as follows.
Point 1: [Abstract] Should be some statement as to signficiance. What is red ginseng and why is it worth looking at?
Response 1: Red ginseng (RG) ingestion has been reported to have effect on body weight, food intake, and fat accumulation reduction. Several studies have reported an association not only between ingestion of Panax ginseng or RG and changes in energy metabolism regulation but also between Panax ginseng or RG ingestion and glycemic control. Also, RG ingestion induced changes in energy metabolism regulation and glycemic control. However, the effect of RG dosage on energy metabolism and the expression of mRNAs related to energy substrate utilization are still unclear. We have raised this issue a little more in detail in the abstract (Lines 12-17).
Point 2: [Line 20] Should say earlier what you are comparing to in terms of excercie - mention previous study.
Response 2: We have included the comparison with previous study in the abstract section (Line 14-16, 25-28)
Point 3: [Lines 20-21] Differed in what way?
Response 3: We have made modifications in detail information in the abstract (Lines 12-15).
Point 4: [Lines 22-23] Needs supportive arguments. If short-term administration had no effects, why bother looking at long term?
Response 2: I agree with you that this sentence is slightly confusing. However, the results in the abstract have been written based on many previous studies mentioned in the discussion.
Previous researches related to the effects of the RG ingestion have been performed for at least 4 to 18 weeks (SJ Park et al. 2013, SH Lee et al., 2012, Oh, Lee, Hwang, & Ji, 2014). Therefore, the effects associated with the ingestion period (long term ingestion) are now being considered. We have made modifications in the article. (Line 28-30)
Point 5: [Lines 34-35] In what organism? Human? Mouse? Where are the references?
Response 2: This sentence describes the sentence that follows, but has provided insufficient explaination. We have deleted that sentence.
Point 6: [Lines 37-38] Need to check causality of the statement. Are you saying that the RG or the high-fat diet improved insulin sensitivity?.
Response 2: The sentence was revised after checking the causality and reference. (Line 43-63)
Point 7: [[Section 2.1] What kind of diet? Standard chow? High-fat?
Response 2: We have added more information about diet in section 2.1. (Line 86-88)
Point 8: [Section 2.2] How were they administered? By gavage? Where the volumes of RG and control similar? (what was the concentration of RG dissolved in water)
Response 2: We have added more information about administration methods and the concentration of RG in section 2.2. (Line 92-95)
Point 9: [Section 2.6] What tissue was used? What were the concentrations of Tag and primers? What were the PCR cycling conditions? What calculation was used - was the RG expressed as fold-change compared to control? Was a house-keeping gene used to normalise expression?
Response 9:
The tissue used for the level of mRNA was gastrocnemius muscles. We have added this information in section 2.6 (Line 144-146). We had made a mistake in describing the initial method of analysis. We used the RT-PCR (reverse transcription PCR) analysis. Thus, we have modified (Section 2.6. Line 143) in the article. In this study, the gene expression of tissue was used as Gastrocnemius muscle. Also, we have set to the PCR conditions of each gene are as follows; GAPDH: 52°C, 18 cycles, GLUT4: 52°C, 23 cycles, MCT1: 52°C, 25 cycles, MCT4: 52°C, 22 cycles, FAT/CD36: 53°C, 24 cycles, CPT1b: 53°C, 24 cycles. RG expressed as fold change was based on CON in percent (%) in figure3. And we expressed the fold change of RG in% based on CON. We normalized using GAPDH as a house-keeping gene.
Point 10: [Table 2] What does the single asterix footnote refer to?
Response 2: We deleted this "* ". This was an incorrect footnote.
Point 11: [Tables 3 and 4] Is only the one value shown as mean ± SE?
Response 2: We removed the single asterisk from Table 4 and 5. There was no significant difference in blood parameters and glycogen concentration between groups.
Point 12: [Line 163] Oxygen uptake is not reported in the results
Response 2: We have additionally inserted oxygen uptake and RMR data. (Section 3.2, Figure 1.)
Point 13: [Line 185] Did you measure leptin?
Response 2: Unfortunately, we did not measure the leptin level. Therefore, we corrected the sentence in the discussion section. (Line 272 - 275)
Point 14: [[Line 190-191] what time of day/night was the RG administered in this study?
Response 2: We administered RG orally daily at 09:00 am during the experiment periods.
We calculated energy expenditure (kcal) of mice active period using the dark phase of 12h and light period (day time) of 12h, respectively. However, there was no significant difference between the two groups in RMR and the active period after RG intake. (added table 3, Line 196-200)
Point 15: [Line 198-199] Oxidation was not reported before the 2 weeks.
Response 2: We have additionally inserted oxygen uptake and RMR data. (Section 3.2, Figure 1.)

Round 2
Reviewer 2 Report
The authors have considered most of the concerns of the reviewer. However, I have still some problems regarding the calculation and interpretation of the calorimetric data. The authors took now the average value of energy expenditure during the light on phase as resting metabolic rate. This is not correct because during this phase there is still some postprandial thermogenesis and also some spontaneous activity. If the authors just want to look at differences in energy expenditute between the light on (rather low activity) and light off (rather high activity) phase, then they should give the results for total energy expenditure separately for the light on and light off phase, respectively. Otherwise, RMR should be calculated as suggested in my first review. Methods and Results chapters have to be adapted accordingly.
Minor: The paper needs some improvement in grammar and style, specifically within the yellow marked (revised) text. In the title, the word "resting" should be deleted.
Author Response
Response to Reviewer 2 Comments
The authors are grateful to Reviewer #2 for valuable comments and suggestions. We revised the manuscript fundamentally in full accordance with the reviewer’s comments: Our responses to each comment are as follows.
Point 1: The authors have considered most of the concerns of the reviewer. However, I have still some problems regarding the calculation and interpretation of the calorimetric data. The authors took now the average value of energy expenditure during the light on phase as resting metabolic rate. This is not correct because during this phase there is still some postprandial thermogenesis and also some spontaneous activity. If the authors just want to look at differences in energy expenditute between the light on (rather low activity) and light off (rather high activity) phase, then they should give the results for total energy expenditure separately for the light on and light off phase, respectively. Otherwise, RMR should be calculated as suggested in my first review. Methods and Results chapters have to be adapted accordingly.
Response 1:
We have recalculated the RMR based on your feedback (first review). (see Table 3). The extraction point for estimating RMR was set to one hour after the light was turned on in the animal metabolic chamber for the mice resting conditions. Also, mice active and inactive period was 12 hours according to the light cycle. The RMR tended to be lower in the RG group, but there was no statistically significant difference between the two groups. (p = .382)
Point 2: The paper needs some improvement in grammar and style, specifically within the yellow marked (revised) text. In the title, the word "resting" should be deleted.
Response 2:
We had deleted "resting" from the title (Line 3). Also, Also, we have a recheck in grammar and style from all revised sentences.

Reviewer 3 Report
I thank the authors for their amendments, however I still have some comments/suggestions as below.
I understand that you looked at a 2wk intervention in your previous exercise study as well, but it seems to be a very short exposure period and may explain why most of the outcomes were not significantly different. Of the other studies menioned in the discussion there seemed to be a trend where the longer studies showed interesting results, while the shorter (4wk, 20 days) did not; however the study length was not mentioned for all previous work. Maybe you should include your original rationale behind the choice of intervention period in the introduction?
[Lines 24-25, 177, 181 266] Gene names should be italicised, otherwise common practice would imply you are referring to protein expression.
[Lines 28-30] Consider rephrasing the last sentence. I'm not sure what you are trying to say.
[Line 64] Why did the mice need to be single-housed?
[Section 2.3] It is not clear that RMR and energy expenditure were only meausred post-2wk.
[Section 2.6] The qRT-PCR cycling conditions and normalisation are still not mentioned.
[Section 3.2] Did you compare only between treatment groups, or also between time points?
[Lines 277-285] There is a lot of repetition in this paragraph.
Author Response
May 28th, 2020
Prof. Dr. Lluis Serra-Majem &. Prof. Dr. Maria Luz Fernandez
Nutrients
Dear Editor:
We wish to re-submit the manuscript titled “
We thank you and the reviewers for your thoughtful suggestions and insights. The manuscript has benefited from these insightful suggestions. I look forward to working with you and the reviewers to move this manuscript closer to publication in the Nutrients.
The manuscript has been rechecked and the necessary changes have been made in accordance with the reviewers’ suggestions. The responses to all comments have been prepared and attached herewith.
Thank you for your consideration. I look forward to hearing from you.
Sincerely,
Kiwon Lim, PhD.
Laboratory of Exercise Nutrition, Konkuk University
120, Neungdong-ro, Gwangin-gu, Seoul 143-701, Republic of Korea
Tel: +82-10-3336-0625; Fax +82-452-6027
E-mail: exercise@konkuk.ac.kr
Response to Reviewer 3 Comments
The authors are grateful to Reviewer #3 for valuable comments and suggestions. We revised the manuscript fundamentally in full accordance with the reviewer’s comments. Our responses to each comment are as follows.
Point 1: I understand that you looked at a 2wk intervention in your previous exercise study as well, but it seems to be a very short exposure period and may explain why most of the outcomes were not significantly different. Of the other studies menioned in the discussion there seemed to be a trend where the longer studies showed interesting results, while the shorter (4wk, 20 days) did not; however the study length was not mentioned for all previous work. Maybe you should include your original rationale behind the choice of intervention period in the introduction?
Response 1:
The endurance training effect in the animal trial previously reported required at least 2–4 weeks. Therefore, in our previous study, we used a minimum of 2-week training periods with the ingestion RG. If endurance training lasted for more than 2 weeks, it would be difficult to distinguish the effects of training from those of RG ingestion. Our previous study determined that 2 weeks of RG ingestion combined with endurance training could increase fat oxidation and exert a glycogen-sparing effect during exercise. Therefore, we wanted to investigate the effect of RG at the same dosage for 2 weeks using untrained mice. We have revised the sentence in the introduction section accordingly (see line 135–138).
[Lines 24-25, 177, 181 266] Gene names should be italicised, otherwise common practice would imply you are referring to protein expression.
Response 1:
We have now provided gene names in italics (Please see line 22–25, 294-296, Table 1, line 306, Figure 3, and line 708-721).
Point 2: [Lines 28-30] Consider rephrasing the last sentence. I'm not sure what you are trying to say.
Response 2:
We have modified this sentence accordingly for clarity (see line 24–27).
Point 3: [Line 64] Why did the mice need to be single-housed?
Response 3:
It was determined that group-housed mice have high stress levels. We thus housed the mice individually to minimize the impact of group housing on energy expenditure. Moreover, we used this strategy to accurately measure the amount of food intake per mouse.
Point 4: [Section 2.3] It is not clear that RMR and energy expenditure were only meausred post-2wk.
Response 4:
This sentence was revised accordingly (please see section 2.3. line 167-168).
Point 5: [Section 2.6] The qRT-PCR cycling conditions and normalisation are still not mentioned.
Response 5:
We have added more detail regarding qRT-PCR cycling conditions and normalization to section 2.6. (see line 294-298).
Point 6: [Section 3.2] Did you compare only between treatment groups, or also between time points?
Response 6:
We measured energy metabolism for 24 h before and after the 2-week experiment. Figure 1 presents the sum of oxygen intake and carbon dioxide production for 24 h. Therefore, the comparison was made between groups. Moreover, the time point comparison was between the active period and inactive period over 24 h, which was divided as indicated in ‘Table 3’ for comparisons between groups.
Point 7: [Lines 277-285] There is a lot of repetition in this paragraph.
Response 7:
We have modified this paragraph to minimize repetition (see line 722-727).
